# Comparison between Sandblasted Acid-Etched and Oxidized Titanium Dental Implants: In Vivo Study

**DOI:** 10.3390/ijms20133267

**Published:** 2019-07-03

**Authors:** Eugenio Velasco-Ortega, Ivan Ortiz-García, Alvaro Jiménez-Guerra, Loreto Monsalve-Guil, Fernando Muñoz-Guzón, Roman A. Perez, F. Javier Gil

**Affiliations:** 1Faculty of Dentistry. University of Seville, 41009 Seville, Spain; 2University of Santiago de Compostela, 27002 Lugo, Spain; 3Bioengineering Institute of Technology, Faculty of Dentistry. Universitat Internacional de Catalunya, Sant Cugat del Vallé, 08198 Barcelona, Spain

**Keywords:** osseointegration, titanium, implant surface, sandblasted surface, acid-etched surface, oxidized surface

## Abstract

The surface modifications of titanium dental implants play important roles in the enhancement of osseointegration. The objective of the present study was to test two different implant surface treatments on a rabbit model to investigate the osseointegration. The tested surfaces were: a) acid-etched surface with sandblasting treatment (SA) and b) an oxidized implant surface (OS). The roughness was measured by an interferometeric microscope with white light and the residual stress of the surfaces was measured with X-ray residual stress Bragg–Bentano diffraction. Six New Zealand white rabbits were used for the in vivo study. Implants with the two different surfaces (SA and OS) were inserted in the femoral bone. After 12 weeks of implantation, histological and histomorphometric analyses of the blocks containing the implants and the surrounding bone were performed. All the implants were correctly implanted and no signs of infection were observed. SA and OS surfaces were both surrounded by newly formed trabeculae. Histomorphometric analysis revealed that the bone–implant contact % (BIC) was higher around the SA implants (53.49 ± 8.46) than around the OS implants (50.94 ± 16.42), although there were no significant statistical differences among them. Both implant surfaces (SA and OS) demonstrated a good bone response with significant amounts of newly formed bone along the implant surface after 12 weeks of implantation. These results confirmed the importance of the topography and physico–chemical properties of dental implants in the osseointegration.

## 1. Introduction

Dental implants have generally shown long-term satisfactory outcomes to replace missing teeth [1]. Titanium and Ti-6Al-4V are widely used in dentistry for different prosthetic rehabilitations [1,2]. Titanium is considered to be an excellent material due to its mechanical strength and elasticity, low density, and stability, among others. Furthermore, titanium presents an excellent corrosion resistance and very low ion release into the physiological medium. In addition, titanium implants have shown to be very stable in vivo and biocompatible, which has been previously shown to favor the osseointegration [3]. The long-term success of dental implants depends on the successful initial osseointegration shortly after implantation. Osseointegration is influenced by various factors, which greatly depend on the host bone and the implant. On the one hand, the osseointegration will depend on the volume and the bone quality at the implant interface, and on the other hand, it will depend on the implant design and the surface treatment present [4,5].

Osseointegration around titanium implants is a complex biological phenomenon not yet clearly understood. Surface modifications of implants, which are mainly centered on the adjustment of the roughness and the compressive residual stresses, as well as on the modification of the chemical compounds, are generally performed in order to increase the appropriate biological response between the living tissues and the dental implants. Overall, this has direct implications in accelerating and improving the osseointegration [6,7]. Furthermore, aside of improving osseointegration, these implant surface modifications have been shown to increase cell viability and biocompatibility [8].

The main modification applied on an implant surface is the roughness; this topographical change is achieved by acid treatments, sandblasting, or different mechanisms of oxidization [9]. The most common process of oxidation is the anodization, which modifies the chemical composition, the microstructure, and the crystal structure [10]. In animal studies, the oxidized surface (OS) implant has been shown to exhibit osteoconductive properties [11]. Oxidized titanium implants produced by electrochemical spark anodization produce a moderately rough surface that increases the rate of osseointegration and the bone contact with the implant by stimulating during the early healing phase a direct bone growth [12]. Sandblasted acid-etched surface (SA) combines sandblasting with abrasive particles and acid etching in order to obtain macroroughness (sandblasting) and micropits (acid etching) in order to produce an optimal roughness, which is expected to have excellent osseointegration [7,9]. SA surface provides an appropriate space for osteoblast adhesion, proliferation, and differentiation [13]. Moreover, surface implant treated with blasting followed by double acid etching produces better osseointegration during the healing phase due to the increase in the specific surface area, which increases the surface available for new bone ingrowth, hence greatly improving the mechanical fixation [14]. The purpose of this study was to test two different implant treatment surfaces on a rabbit animal model to investigate the osseointegration through histological and histomorphometric analysis.

## 2. Results

### 2.1. Surface Characterization

Scanning electronic microscopy (SEM) showed two substantially different types of topographies (Figure 1). The OS surface exhibited a porous morphology with many pores with elevated margins that resembled volcanoes that were produced by the oxidation manufacturing process. SA surfaces showed a rough topography characterized by irregular cavities homogeneously alternated with peaks and valleys (Figure 1).

Roughness analysis revealed the main differences between SA and OS surfaces in terms of the general topographic parameters, mainly Ra (the average roughness), which refers to the arithmetic average of the absolute values of the distance of all points of the profile to the mean line, and Rt, which expresses the distances between the highest points (valleys). In particular, the OS surface exhibited the lowest roughness values (Ra 1.37 μm, Rt 24.21 μm) compared to the SA (Ra 1.76 μm, Rt 31.37 μm). The parameter Rq, which aims at describing the irregularity distribution, was slightly higher for the SA surface compared to the OS surface (2.37 μm versus 1.50 μm). The parameter Rz, which expresses the differences between the highest and the lowest points, was slightly higher for the SA surface compared to the OS surface (22.80 μm versus 20.77 μm). These data demonstrated a higher roughness on the surface of the implants treated with sandblasted and etching (SA) than on the oxidized surfaces (OS), showing statistically significant differences between the two types of surfaces (*p* < 0.005) (Table 1). 

The results of the residual stresses presented a compressive character (negative values) due to the plastic deformation caused by the projection of abrasive particles at a pressure of 2.5 MPa. Residual stresses are presented in Table 1. The compressive stresses induced by oxidized surfaces were lower than those induced by sandblasting, presenting statistically significant differences among them (*p* < 0.001, t-Student).

### 2.2. Histological and Histomorhometric Study

All in vivo experiments did not present complications, showing normal healing after the surgeries. At the time of sacrifice, all dental implants were submerged and covered by a healthy ridge of skin. All implants were in situ when animals were sacrificed. At retrieval, a macroscopic exam of the implant site was performed, which demonstrated that all the implants had been correctly inserted and no signs of infection were observed. A good biological response was observed on the two implant surfaces (SA and OS), showing an important amount of new bone formation after 12 weeks of healing. The new bone formed presented activity, histologically revealing very similar bone to the old bone tissue. SA and OS implant surfaces were surrounded by newly formed trabeculae of woven bone as can be observed in Figure 2.

Although histomorphometric analysis revealed that the bone–implant contact (BIC) was higher around the SA implants than around the OS implants, there were no significant differences between the two types of surfaces (*p >* 0.005.; Table 2). 

Histological analysis was performed to characterize the different implant regions. Bone volume per tissue volume (BV/TV) was determined on the cervical, medial, and apical thirds of the implant. The histomorphometric findings obtained from light microscopy showed in both implant surfaces a greater BV/TV in the cervical region and a lower BV/TV in the medial region. (Table 3).

## 3. Discussion

Osseointegration of titanium dental implants is a complex phenomenon depending on different factors and characterized by a long sequence of biologic events: wettability, unspecific protein adsorption and migration, proliferation and differentiation of cells, the induction of genes related to growth, maturation and organization of new bone, and finally, a matrix mineralization phase [15]. Osseointegration can be influenced by the nature of the material of the implant, mechanical characteristics, biocompatibility, surface and design of the implant, the bone quantity and quality, load applied to the implant, and even the surgical technique used for implantation [16]. Due to the large number of variables involved in the success of osseointegration, experimental and clinical studies should be conducted to clarify the role and the importance of each variable in the success of the implant. Several animals such as rabbits are commonly used, due to the similar mechanical properties of the human bone, as subjects for screening dental implant materials. They have gained great attention for their numerous advantages even though larger animal models should be ideally used [17]. 

Different treatments of titanium surface can modify the physico–chemical behavior and microstructural properties of the implant that in turn are able to affect bone formation processes: sandblasting, acid etching, physical or chemical vapor deposition, spark anodization, oxidation, laser treatments, or cold gas spray, among others. The most commonly applied treatments are the sandblasting combined with acid etching and oxidation processes by electrochemical treatments. Surface properties may play a critical role in biomolecular adsorption and cell adhesion to the implant surface as well as in osteoblast cell maturation. However, the selection of an optimal surface topography and roughness is still not clear and is hence a controversial issue [16]. A recent study analyzed the influence of implant surface by using four different surfaces in a rabbit tibiae model, comparing blasted, acid-etched, and discrete hydroxyapatite deposition; blasted; acid-etched; and blasted and acid-etched [18]. The authors clearly observed better results in blasted–etched and covered with hydroxyapatite surfaces, although no statistical differences were obtained in the study [18]. The implants were mainly in contact with cortical bone along the upper threads, while the threads in the bone marrow were in contact with either newly formed bone or normal marrow tissue [18]. 

Roughness and physico–chemical properties, especially residual stresses, are the most relevant characteristics of the surface properties for clinical success. For this reason, the present study analyzed the surface roughness and residual stresses of the implants. The results of this study showed that SA surface was rougher than OS surface according to Ra, Rt, Rq, and Rz parameters (Table 1). The topography is known to affect the protein adsorption and cellular interaction with the biological environment, which influences the orientation, adhesion, migration, proliferation, growth, and differentiation of osteoblastic cells. In vitro and in vivo tests have shown that surface roughness improves the osseointegration of titanium implants as long as the R_a_ values are in the range of 1.5 to 3.5 micrometers [8,19]. The formation of bone on titanium interfaces depends on the direct interactions of osteoblasts and the formation of osteocalcin and apatite deposition of bone matrix. Therefore, the proper osteoblast adhesion and the formation of osteoblast extracellular matrix are the main steps for the successful osseointegration of implants [20]. Human cells cultured in treated titanium surfaces (i.e., sandblasting, acid etching) showed more signs of cell differentiation as compared to machined surfaces. In fact, a recent study has demonstrated that osteoblasts cultured on rougher surfaces tended to exhibit properties of more differentiated osteoblasts than those cultured on smoother surfaces [8]. 

It is well known that the sand projection on titanium surfaces increases the implant surface roughness depending on the hardness, size, and nature of the abrasive particle as well as on the pressure and distance from the projection gun. The impingement produced, at high pressure, produces local plastic strain and compressive residual stresses. This fact produces an increase in osteoblast adhesion as well as earlier fixation and better osseointegration of the dental implant as has been previously described by Aparicio and Gil [21,22]. The compressive stress produces an increment in the adhesion, proliferation, and differentiation of human osteoblast. The residual stress is higher in sandblasted and acid-etched implants in comparison with the oxidized implants. The compression stress is caused in SA by the plastic deformation caused by the abrasive particles’ projection, whereas in the OS implants, due to the significant increase of oxygen atoms in the surface, titanium oxide is produced, which leads to a volume increase, hence producing compressive residual stress.

In this contribution, the bone formation of two types of implant surfaces were compared in the bone of New Zealand rabbits. Overall, the results of this experimental study indicated that the SA surface presented more bone index contact than the oxidized surface, after 12 weeks of healing. A study investigated the effects of surface modifications in an in vivo study by comparing chemically modified hydrophilic sandblasted, large-grit, acid-etched (modSA), and anodically oxidized hydrophobic implant surfaces [19]. Their results suggested that the hydrophilic modSA surface could have a stronger affinity for bone than the oxidized surface during the initial healing period. In fact, the study found blood clots close to the hydrophilic modSA surface [19]. The energy of surface treatment is important for the initial adhesion of proteins—selective adhesion as fibronectine—and may enhance the interaction between titanium and biological environment, improving bone formation and osseointegration of the dental implant [23]. 

In the present study, both implant surfaces (SA and OS) demonstrated a good bone response with an amount of new bone formation along the implant surface after 12 weeks of healing. These results were confirmed in a recent study, demonstrating the importance of the surface physico—chemical properties of the dental implants in their osseointegration [24]. In this previous work, three types of surface on titanium implants (micro SA, nano OS, and machined) were carefully prepared and characterized prior to implantation in rabbit femurs. No significant differences were shown between the different groups regarding the osseointegration [22]. SA and OS surfaces showed good biological properties [13,25]. The human osteoblasts presented a very good adhesion, proliferation, and differentiation on SA surface [13]. In the in vivo study, new bone tissue was formed around the cortical bone of the tibia. Furthermore, the bone regeneration continued along the dental implant to the hollow region of the tibiae, and to the apex of the implant (bone narrow region). This indicated that the sandblasted surface had excellent biocompatibility and bone-forming capacity [13]. Also, OS surface showed an excellent biological response. OS surface increased the biocompatibility and hemocompatibility. The OS surface characteristics have been found to improve cell proliferation, adhesion, and spreading [25]. 

It is generally accepted that in in vivo studies, there are differences between different animals, although in this study, 30 image captures have been analyzed by microscopy. No differences in the BIC percentages were observed between the different regions of the implanted dental implants. We must consider that the study has some limitations such as the differences in the design of the two experimental dental implants and the small differences in the surgical techniques in the implantation. In any case, we can conclude that SA implants have a higher level of osseointegration than OS implants, although the differences are not statistically significant considering *p* < 0.05 but they would be with a confidence of *p* < 0.13.

High levels of osseointegration are important to guarantee the mechanical stability of the dental implant. In this study, we have seen the influence of roughness and residual stress on osseointegration in dental implants with very similar designs. The clinicians should take into account these aspects in the selection of dental implants [26].

## 4. Materials and Methods 

### 4.1. Implants

In the present study, two different types of implants were used: Branemark System Mk III Groovy RP, diameter 4.1 mm, length 10 mm (Nobel Biocare AB, Göteborg, Sweden) with an oxidized implant surface TiUnite™; and Surgimplant, diameter 4 mm, length 10 mm (Galimplant, Lugo, Spain) with a sandblasted acid-etched surface Nanoblast™.

Roughness was evaluated according to the recommendations by Wennerberg and Albrektsson [27] on topographic evaluation for dental implants. An interferometeric microscope (Wyko NT1100, Veeco) was used. The surface analysis area was 227 × 298 µm^2^ for the two surfaces. Data analysis was performed with Wyko Vision 232TM software (Veeco, New York, NY, USA). A Gaussian filter was used to separate waviness and form from the roughness of the surface.

Surface topography was performed by means of scanning electron microscopy (SEM, Philips 515, Philips, Eindhoven, The Netherlands). 

An X-ray diffractometer with Bragg–Bentano configuration (D-500, Siemens, Germany) was used to evaluate the residual stress. The determination was realized for the family of planes (213), which diffracts at 2θ = 139.5°. Titanium elastic constants at the direction of this family of planes are calculated as EC = (E/1 + υ) _(213)_ = 90.3 (1.4) GPa. Eleven Ψ angles, zero and five negative and five positive angles were studied. The position of the peaks was determined with a pseudo-Voigt function using Winplot software, in order to convert to interplanar distances (d Ψ). Calculations were realized following the Bragg’s equation. The d Ψ vs sen^2^Ψ graphs and the calculation of the slope of the linear regression (A) were realized with appropriate software (Origin, Microcal, USA). The residual stress is: σ = EC * (1/d_0_) *A; where d_0_ is the interplanar distance for Ψ = 0°.

### 4.2. Animals

Six adult female New Zealand rabbits (G. San Bernardo, Navarra, Spain) of 6–7 months of age and mean weight 5 kg were used, after approval by the ethical committee of the Facultad de Veterinaria of University of Santiago de Compostela, Spain (Ref. 18/USC/EC/ 35674-approved 19/3/2018). Procedures were conducted in the Rof Codina Clinic of the University of Santiago de Compostela (Lugo, Spain). All experiments were performed according to the Spanish Government Guide and to the European Guide for Animal Care. Animals were housed in cages, allowed to perform a normal activity, and monitored once a day by trained staff to assess changes in general health.

### 4.3. Surgical Procedures

All surgical procedures were done under sterile conditions, in an operating room and under general anaesthesia induced and maintained on a concentration of 2.5–4% of isoflurane (Isoba-vet, Schering-Plough, Madrid, Spain). The animals were first premedicated with a combination of medetomidine (50 µg/kg/i.m., Domtor, Esteve, Barcelona, Spain) and ketamine (25 mg/kg/i.m., Imalgène 1000, Merial, Toulouse, France). During anesthesia, the animals were continuously monitored by a veterinarian category B or C.

Each animal received peri- and postoperative analgesia using buprenorphine (1 mg/Kg/i.m., Buprex, RB Pharmaceuticals, Berkshire, UK), antibiotic prophylaxis during one week with enrofloxacin (15 mg/Kg/s.c. once a day, Ganadexil 5%, Invesa, Barcelona, Spain), and pain controlled with meloxicam (20 μg/Kg/s.c., Metacam, Boehringer Ingelheim, Barcelona, Spain) during three days.

The experimental site was located on both distal lateral condyles of the femur. After three weeks of quarantine, general anesthesia was induced in the animals. After shaving and disinfecting, the femoral condyles were exposed by a lateral longitudinal incision. Implant bed preparations were carried out according to the manufacturers’ guides. Finally, the muscle, the subcutaneous tissue, and the skin were sutured in layers with reabsorbable sutures (Vicryl 4-0, Ethicon, New Jersey, USA).

Twelve weeks later, rabbits were painlessly sacrificed by a sodium pentobarbital overdose (100 mg/Kg/i.v., Dolethal, Vétoquinol, Madrid, Spain) after sedation with ketamine (25 mg/Kg/i.m., Imalgène 1000, Merial, Toulouse, France).

### 4.4. Histological and Histomorphometric Analyses

The blocks containing the implant, distracted bone, and the hard and soft tissues around the dental implant were obtained using an oscillating saw, fixed, and identified. These blocks were dehydrated in different graded ethanol series (70–100%) and infiltrated with different graded mixtures of ethanol and glycometacrylate (Technovit 7200 VLC, Heraus Kulzer, Werheim, Germany) following guidelines previously published [28]. The samples were then polymerized and heated at 37 °C for 24 h to ensure a complete polymerization.

Longitudinal sections (central sections) direction of 200 microns were done of implants using a band saw and mechanically polished (Exakt Apparatebau, Norderstedt, Germany) using 1200 and 4000 grit silicon carbide papers (Struers, Copenhagen, Denmark) until samples were obtained with a thickness of approximately 40 μm. The slides were stained using the Levai–Laczkó method [29] for both histological and histomorphometric analysis.

Quantitative and semiquantitative histology was performed using motorized light microscopy and a digital camera connected to a PC-based image capture system (BX51, DP71, Olympus Corporation, Japan). Peri-implant tissues were the regions of interest. Image analysis was conducted based on color, differentiating the new and lamellar bone from the connective and vascular tissues (Adobe Photoshop, CA, USA). Parameters were evaluated and measured by a masked examiner using PC-based image analysis program Cell-sens 1.5 (Olympus Corporation, Japan). Only one researcher studied the 30 slides (image capture) for each dental implant in order to avoid the possible different interpretations.

Firstly, a semiquantitative histological evaluation was performed according to the ISO 10993-6 protocols. The following grading scale was used: absent, 1 = slight, 2 = moderate, 3 = marked, and 4 = severely irritant.

The implant surface in contact with mineralized bone, referred to as “bone-to-implant contact” (BIC), was calculated as a percentage. “Bone volume per tissue volume” (BV/TV) of total or new bone was determined to within a distance of 300 µm immediately adjacent by dividing the area of bone volume (BV) or new bone volume (nBV) by the total tissue volume (TV) present in the region of interest. BV/TV was determined on the coronal, medial, and apical thirds of the implant.

### 4.5. Statistical Analysis

The animal was chosen as the unit for the statistical analysis. The primary outcome parameters were BIC and nBV/TV at the different compartments. The data were reported by using means, standard deviations (SD), ranges, 95% confidence intervals (CI), and medians (SPSS, SPSS Inc., Chicago, IL, USA). Paired two-sample t test was performed. Results were considered as significant at *p* < 0.05.

## 5. Conclusions

Sandblasted and acid-etched implants (SA) produce a macroroughness with a microroughness on the titanium surface which presents higher bone index contact than the oxidized surface (OS). Residual stresses on the surface are produced by the abrasive particles’ projection at high pressure that induce a compressive stress which favors the osseointegration. Residual stresses on the oxidized titanium surfaces (OS) are lower. After implantation for 12 weeks in rabbit bone, the hard tissue formed and anchorage on the OS surface was slightly lower than on the SA implants. The roughness and residual stress favors the bone integration on the SA surface in relation to the other OS surface.

## Figures and Tables

**Figure 1 ijms-20-03267-f001:**
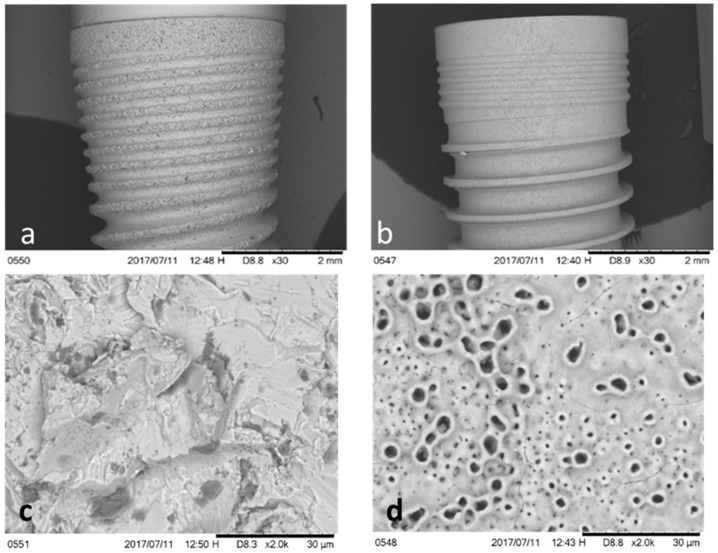
Scanning Electronic Microscope images of the two different implants tested: (**a**) dental implant sandblasted and acid-etched surface (SA); (**b**) dental implants with oxidized surface (OS); (**c**) surface of SA dental implants; (**d**) surface of OS dental implants.

**Figure 2 ijms-20-03267-f002:**
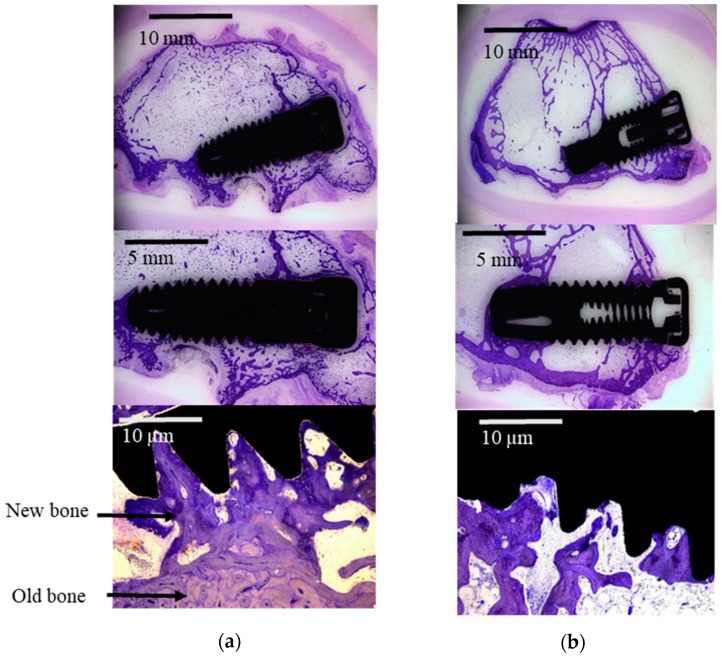
Light microscope images of the two different implants tested: (**a**) sandblasted and acid-etched surface implant (SA); (**b**) oxidized surface implant (OS).

**Table 1 ijms-20-03267-t001:** Surface profile parameters (μm) and residual stress (σ) in MPa.

Parameter	SA	OS	*p*
Ra	1.76 ± 0.21	1.37 ± 0.11	0.0008
Rt	31.37 ± 2.30	24.21± 2.89	0.0002
Rq	2.37± 0.12	1.50 ± 0.13	0.0003
Rz	22.80 ± 3.30	20.77± 4.00	0.0001
Residual stress	−213.3 ± 3.6	−71.0 ± 5.1	0.0001

**Table 2 ijms-20-03267-t002:** % Bone–implant contact.

Animal	SA	OS	*p*
1	62.74 ± 8.01	30.01 ± 15.88	0.02
2	46.04 ± 7.56	71.77 ± 20.22	0.33
3	64.11 ± 8.43	42.80 ± 18.13	0.28
4	47.33 ± 8.67	32.03 ± 14.34	0.04
5	56.14 ± 7.56	41.03 ± 21.10	0.03
6	45.76 ± 10.53	37.98 ± 8.85	0.03
Mean ± SD	53.49 ± 8.46	50.94 ± 16.42	

**Table 3 ijms-20-03267-t003:** Regional distribution of BV/TS.

Region	SA	OS
Cervical	41.22 ± 5.12	38.54 ± 4.56
Medial	29.97 ± 3.21	19.97 ± 3.99
Apical	41.61 ± 4.23	47.67 ± 5.67

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
