# Peer review of "Comparison between Sandblasted Acid-Etched and Oxidized Titanium Dental Implants: In Vivo Study"

_ijms, 2019, doi:10.3390/ijms20133267_

Reviewer 1 Report

The present rabbit study reports on histology and histomorphometry of two rough implant surfaces: sandblasted-acid etched (SA) and oxidized (OS) surfaces. The roughness was not statistically significantly different between the two groups and correlated with a bone-to-implant contact (BIC) that was not statistically significantly different between the two investigated surfaces.

Comments

Introduction:

Please modify or remove the sentence “Dental implants are the best solution in order to replace missing teeth”. Indeed, dental implants are not always the best solution and there are cases most suitable for other treatments like bridges or removable prostheses. Possible alternatives for the sentence are “Dental implants showed long term satisfactory outcomes to replace missing teeth” or “Dental implants are documented and predictable solution in order to replace missing teeth”. In case the first sentence will be modified and not removed, the following article could be used as supporting citation: Berglundh et al. J Clin Periodontol 2002; 29(Suppl. 3): 197–212.

Results:

·         Please provide P-value for each roughness parameter between OS and SA.

·         The manuscript refers to “Figure 3-5”, but the figure’s legend is named as Figure 3 only. Please include Figure 4 and Figure 5, or modify the manuscript accordingly.

Discussion:

·         The Discussion reports “a greater roughness in SA surface implants than OS surface” but in the Results there is no statistically significant difference (“there was no significant difference between the two types of surface implants (P>0.05)”). Please do not stress differences if they are not at a level of statistical significance.

·         “SA surface presented more bone index contact than the oxidized surface, after 12 weeks of healing” but in the results the P-value for BIC is > 0.05. As mentioned above, it is not recommended to stress differences if they are not statistically significant.

·         Please consider reducing or summarizing the proposed Discussions and including a small paragraph on clinical implications for the present study.

·         Please add a small paragraph on the limitations of the study.

·         Please suggest future directions for articles on the same topic. For example, future articles could present histology at different timepoints to have longitudinal documentation of osseointegration. A classical citation for longitudinal documentation of osseointegration is Lang et al. Clin. Oral Impl. Res. 22, 2011; 349–356.

Conclusions:

·         As proposed for the Discussions, it is not recommended to underline differences if not statistically significant.

Additional comments:

·         Macrodesign of the two investigated implants was not compared. From Figure 2 it is notable that both implants have micro-threads at the implant neck, but it seems that they differ for thread pitch and design. A discussion that could add value to the article is that, despite different macrodesigns, similar value of roughness correlated with similar BIC. A possible clinical implication for the mentioned finding is that the effect of macro-design (crestal module, micro-threads at the implant neck…) become most relevant after the implant uncovering, while its effect is negligible during the process of osseointegration.

·         The above-mentioned discussion could also be applied to Figure 1. Indeed, despite dissimilar appearance for surface topography at the Scanning Electronic Microscope, similar value of roughness correlated with similar BIC. It suggests that osseointegration is conditioned by surface roughness and not by topography. Future studies comparing surfaces with similar roughness but different topography should also include evaluation of gene expression.

Language review:

Please consider contacting a native English speaker for an extensive language editing.

The article presents with typing mistakes. In the Abstract there is “streess Bragg-Bentano diffraction”, please remove one “e” from streess. In the Results there is a parenthesis opened and never closed: “(SA than oxidized surface (OS) implants”.

Author Response

The authors thank the reviewer for their comments and suggestions. We have taken into account all the comments and we have corrected everything. Thanks to these comments, the manuscript has improved in quality and clarity.

Please modify or remove the sentence “Dental implants are the best solution in order to replace missing teeth”. Indeed, dental implants are not always the best solution and there are cases most suitable for other treatments like bridges or removable prostheses. Possible alternatives for the sentence are “Dental implants showed long term satisfactory outcomes to replace missing teeth” or “Dental implants are documented and predictable solution in order to replace missing teeth”. In case the first sentence will be modified and not removed, the following article could be used as supporting citation: Berglundh et al. J Clin Periodontol 2002; 29(Suppl. 3): 197–212.

DONE. The reference has been icnorporated.

Please add a small paragraph on the limitations of the study.

DONE (in discussion section)

The article presents with typing mistakes. In the Abstract there is “streess Bragg-Bentano diffraction”, please remove one “e” from streess. In the Results there is a parenthesis opened and never closed: “(SA than oxidized surface (OS) implants”.

DONE

The manuscript refers to “Figure 3-5”, but the figure’s legend is named as Figure 3 only. Please include Figure 4 and Figure 5, or modify the manuscript accordingly.

DONE. The figure has been improved and we have added more information.

Please provide P-value for each roughness parameter between OS and SA.

DONE. p-values have been incorporated for each roughness parameter and BIC %.

 As proposed for the Discussions, it is not recommended to underline differences if not statistically significant.

DONE. In conclusions section.

The Discussion reports “a greater roughness in SA surface implants than OS surface” but in the Results there is no statistically significant difference (“there was no significant difference between the two types of surface implants (P>0.05)”). Please do not stress differences if they are not at a level of statistical significance.  “SA surface presented more bone index contact than the oxidized surface, after 12 weeks of healing” but in the results the P-value for BIC is > 0.05. As mentioned above, it is not recommended to stress differences if they are not statistically significant.

DONE. Roughness and residual stresses between the two surfaces studied present statistical differences p<0.05. The p values have been incorporated in the text and has been crrected the mistake in the text. Roughness presents statitiscal differences. The p-values have been incorporated in the Tables 1 and 2

Please consider reducing or summarizing the proposed Discussions and including a small paragraph on clinical implications for the present study.

DONE. The reference has been incoporated.

Additional comments. 

DONE. The comments very appropriated of the reviewer have been taken into account and in different paragraphs have been incorporated. In this sense, the text is more clear.

Reviewer 2 Report

Thank you for this interesting manuscript. This manuscript delivers insight into the ossification potential of two different treated surfaces of titanium dental implants. 

Nevertheless, the manuscript needs some revision to improve the validity of this study.

The language and mathematic punctuation should be revised carefully. There are a lot of slips of the pen. For example on page 1 in abstract row 3 “rabitt” and row 6 “streess”. Also, commas instead of dots are used, as a consequence the described implant has a diameter of 4100 mm (page 7; 2.1 implants …diameter 4,1 mm,…). Numeration of chapter should be adapted.

Figure 1 & Figure 2:

It seems that legends of figures are interchanged. Scale bar in figure 1 has a range of x2000 magnification and legend of figure 2 repeats it.  For a better comprehensibleness, figures should be summarized in one figure.

Results:

3.1 Surface:

Page 3: Parameter Ra and Rt should be described in the same way as parameter Rq and Rz. Not all readers are familiar with the parameter. Values of t-student test should be added in table 1.

3.2 Histological and histomorphometric study:

Statement of formation of new bone is with the given figures not possible. Quality of figures is bad. It is not possible to differ between old and new bone. Maybe it will be clearer when arrows lead the view. Further, samples/slides of (b)-row are not identical. For a better reliability, zoom of the same slides should be used. Presentation of scale bars is puzzling. Scale bars should be added directly in the picture and as an inherent part. In this way, how is secured that the scale bar is altered in the same way as the figure, when figures are maybe zoomed for a better presentation?  Magnification should be added to figure legends 3.

Authors differs between SA (sandblasted and acid etched) and OS (oxidized) surfaces. On page 5, authors used for OS ”…anodized implants…”. Authors should use the same denomination, although both descriptions are correct. 

Authors suggest that the bone-implant contact (BIC) is higher around the SA implants. This is correct, when mean values are used. On the other hand, looking at each animal in 3 of 6 animals SA as well as OS is higher. Authors should discuss this. Is there a reason for this mismatch? In table 3, authors differ between regions of the implants. Data for each animal should be added, likewise table 2. For the BIC, there are strong discrepancies between each animal and the mean value. Maybe this is also between the regions? If yes, study design should be revised. If not, data could be reanalysed with new perceptions.

Materials and Methods:

2.2 Animals

Number of ethic approval should be added.

2.4 Histological and histomorphometric analyses

Page 9: Number of slides per animal (n=?) to gain semiquantitative score should be added. And number of persons, who determine score, should be added.

Author Response

The authors thank the reviewer for comments that improve the quality of the manuscript. We have made all the corrections indicated:

Figure 1 & Figure 2:

It seems that legends of figures are interchanged. Scale bar in figure 1 has a range of x2000 magnification and legend of figure 2 repeats it.  For a better comprehensibleness, figures should be summarized in one figure.

DONE

2.2 Animals

Number of ethic approval should be added.

DONE

2.4 Histological and histomorphometric analyses

Page 9: Number of slides per animal (n=?) to gain semiquantitative score should be added. And number of persons, who determine score, should be added. 

THESE INFORMATIONS HAVE BEEN INCORPORATED. DONE

Authors differs between SA (sandblasted and acid etched) and OS (oxidized) surfaces. On page 5, authors used for OS ”…anodized implants…”. Authors should use the same denomination, although both descriptions are correct. 

DONE

The language and mathematic punctuation should be revised carefully. There are a lot of slips of the pen. For example on page 1 in abstract row 3 “rabitt” and row 6 “streess”. Also, commas instead of dots are used, as a consequence the described implant has a diameter of 4100 mm (page 7; 2.1 implants …diameter 4,1 mm,…). Numeration of chapter should be adapted.

REVISED

Page 3: Parameter Ra and Rt should be described in the same way as parameter Rq and Rz. Not all readers are familiar with the parameter

DONE

Statement of formation of new bone is with the given figures not possible. Quality of figures is bad. It is not possible to differ between old and new bone. Maybe it will be clearer when arrows lead the view. Further, samples/slides of (b)-row are not identical. For a better reliability, zoom of the same slides should be used. Presentation of scale bars is puzzling. Scale bars should be added directly in the picture and as an inherent part. In this way, how is secured that the scale bar is altered in the same way as the figure, when figures are maybe zoomed for a better presentation?  Magnification should be added to figure legends 3.

DONE. 

Authors suggest that the bone-implant contact (BIC) is higher around the SA implants. This is correct, when mean values are used. On the other hand, looking at each animal in 3 of 6 animals SA as well as OS is higher. Authors should discuss this. Is there a reason for this mismatch? In table 3, authors differ between regions of the implants. Data for each animal should be added, likewise table 2. For the BIC, there are strong discrepancies between each animal and the mean value. Maybe this is also between the regions? If yes, study design should be revised. If not, data could be reanalysed with new perceptions.

THESE ASPECT HAVE BEEN CONSIDERED IN THE DISCUSSION OF THE RESULTS. THE AUTHORS HAVE INTRODUCED IN THE TEXT, THE LIMITATIONS OF THE STUDY,  AND THE LEVEL OF STATITICAL DIFFERENCES.

Numeration of chapter should be adapted.

DONE

The language and mathematic punctuation should be revised carefully. There are a lot of slips of the pen. For example on page 1 in abstract row 3 “rabitt” and row 6 “streess”

REVISED

Thank you very much for your attention and consideration

Round  2

Reviewer 1 Report

General comments:

The revised manuscript improved from its first version. However, the quality of presentation still needs to be improved.

Specific comments:

Results:

1.       “there was statistical significant difference between the two types of surface implants (P>0.05) (Table 1)”. If the difference is statistically significant, the P-value should be < 0.05. Indeed, Table 1 shows all P-values reporting statistically significant differences.

2.       Table 2. “Media ± SD”. Please translate Media ± SD with Mean ± SD.

Extensive English editing is needed. The authors are recommended to contact a native English speaker with experience in scientific writing to improve the language of the manuscript.

Author Response

Thank you very much for the new comments.

The authors have corrected the mistake p<0.005 instead of p<0.005 and mean instead of media.

The English has been revised by a native professor of the University.

Best wishes

FJ Gil